# Validation of a Quantitative Proton Nuclear Magnetic Resonance Spectroscopic Screening Method for Coffee Quality and Authenticity (NMR Coffee Screener)

**DOI:** 10.3390/foods9010047

**Published:** 2020-01-04

**Authors:** Alex O. Okaru, Andreas Scharinger, Tabata Rajcic de Rezende, Jan Teipel, Thomas Kuballa, Stephan G. Walch, Dirk W. Lachenmeier

**Affiliations:** 1Department of Pharmaceutical Chemistry, University of Nairobi, P.O. Box 19676-00202 Nairobi, Kenya; alex.okaru@gmail.com; 2Chemisches und Veterinäruntersuchungsamt (CVUA) Karlsruhe, Weissenburger Straße 3, 76187 Karlsruhe, Germany; andreas.scharinger@cvuaka.bwl.de (A.S.); tabata.rajcicderezende@cvuaka.bwl.de (T.R.d.R.); jan.teipel@cvuaka.bwl.de (J.T.); thomas.kuballa@cvuaka.bwl.de (T.K.); stephan.walch@cvuaka.bwl.de (S.G.W.)

**Keywords:** caffeine, 16-*O*-methylcafestol, kahweol, furfuryl alcohol, tetramethylsilane (TMS), magnetic resonance spectroscopy, validation studies

## Abstract

Monitoring coffee quality as a means of detecting and preventing economically motivated fraud is an important aspect of international commerce today. Therefore, there is a compelling need for rapid high throughput validated analytical techniques such as quantitative proton nuclear magnetic resonance (NMR) spectroscopy for screening and authenticity testing. For this reason, we sought to validate an ^1^H NMR spectroscopic method for the routine screening of coffee for quality and authenticity. A factorial experimental design was used to investigate the influence of the NMR device, extraction time, and nature of coffee on the content of caffeine, 16-*O*-methylcafestol (OMC), kahweol, furfuryl alcohol, and 5-hydroxymethylfurfural (HMF) in coffee. The method was successfully validated for specificity, selectivity, sensitivity, and linearity of detector response. The proposed method produced satisfactory precision for all analytes in roasted coffee, except for kahweol in canephora (robusta) coffee. The proposed validated method may be used for routine screening of roasted coffee for quality and authenticity control (i.e., arabica/robusta discrimination), as its applicability was demonstrated during the recent OPSON VIII Europol-Interpol operation on coffee fraud control.

## 1. Introduction

Coffee remains a popular beverage worldwide and is typically obtained from the two species, namely *Coffea canephora* (robusta) and *Coffea arabica* [1,2,3]. *Coffea arabica* fetches a higher price in the market owing to its perceived superior organoleptic properties and higher production costs compared to *Coffea canephora* [4]. Consequently, beverage fraud involving complete or partial substitution of arabica with robusta coffee cannot be overruled. On the other hand, the diterpenes cafestol, 16-*O*-methylcafestol (OMC), and kahweol found in the lipid fraction of coffee serve as potential markers for the differentiation of *C. canephora* and *C. arabica*. Cafestol is found in both *C. canephora* and *C. arabica* while OMC is specific only to *C. canephora* [5,6,7]. Kahweol, although present in both types of coffee, is found in significantly higher amounts in *C. arabica*. These differences in the diterpene constituents enable the distinction between the coffees and also enable detection of beverage fraud involving substitution of *C. arabica* with the cheaper *C. canephora* beans using OMC as a marker [8].

A number of analytical techniques such as high performance liquid chromatography (HPLC) [9], gas chromatography with flame ionization detection [10], gas chromatography-mass spectrometry [11,12], proton transfer mass spectrometry [13], nuclear magnetic resonance (NMR) spectroscopy [14], isotope-ratio mass spectrometry [15], near-infrared spectroscopy [16,17], electronic nose [18], flame atomic absorption spectrometry [19], and attenuated total reflectance Fourier transform infrared spectroscopy [16], among other techniques, has been reported in the literature for the quantitative determination of coffee constituents and screening of coffee for adulteration.

Nuclear magnetic resonance spectroscopy in combination with chemometrics has been applied either for routine quality control and/or detection of potentially harmful substances in beverages such as alcohol [20], fruit juices [21,22] and coffee [8,23]. NMR spectroscopy may be applied for the quantification of caffeine, OMC, kahweol, 5-hydroxymethylfurfural (HMF) and furfuryl alcohol in coffee [24,25,26]. For decaffeinated coffee, NMR spectroscopy may be used to determine the residual quantities of caffeine, which would typically be less than 1 g/kg. Furfuryl alcohol and HMF may be used as indicators of the degree of coffee roasting [24]. However, furfuryl alcohol is also of public health significance and therefore may require monitoring using NMR. The International Agency for Research on Cancer (IARC) classifies furfuryl alcohol into Group 2B (possibly carcinogenic) [27]. NMR also offers the advantages of cost-effectiveness especially for screening. Additionally, NMR provides reproducible quantitative data [28,29] and generates unique chemical fingerprints that may be useful for authenticity testing [30,31]. Similar to other analytical techniques, reliable results may only be obtained by use of validated methods. Based on previously published method development and optimization work [14,24,25,26], the aim of this study was to validate the quantitative NMR spectroscopic method for screening coffee for both quality and authenticity.

## 2. Materials and Methods

### 2.1. Chemicals

Reagents and standard compounds were of analytical or HPLC grade. The five analytes caffeine, HMF, OMC, kahweol and furfuryl alcohol were purchased from Sigma-Aldrich (Steinheim, Germany). Deuterated chloroform-d_1_ (≥99.8% atom % D) and internal reference standard tetramethylsilane (TMS) were obtained from Roth (Karlsruhe, Germany).

#### Reference Standards and Preparation of Working Standard Solutions

Primary stock solutions of caffeine, HMF, OMC, kahweol, and furfuryl alcohol were prepared in deuterated chloroform solution with 1% TMS (CDCl_3_ + TMS). Individual stock solutions were prepared by separately dissolving 5 mg of caffeine, HMF, kahweol and furfuryl alcohol each in 5 mL CDCl_3_ + TMS. For preparation of OMC stock solution, 10.9 mg of OMC powder was dissolved in 10.9 mL CDCl_3_ + TMS. Working solutions were obtained by carrying out a 1:2 dilution. The stock solutions were kept in the freezer until use. The guidance concentration and defined working ranges for the working standards are given in Table 1. A control solution was prepared by dissolving 25.02 mg 1,2,4,5-tetrachloro-3-nitrobenzene with CDCl_3_ to 5 mL.

### 2.2. Methodology

#### 2.2.1. Samples and Sample Preparation

Coffee samples (commercial products from local supermarket in Karlsruhe, Germany) for analysis were prepared by weighing 200 mg of ground beans before being dissolved in 1.5 mL of CDCl_3_ + TMS. The samples were shaken for 10 min or 20 min at 350 rpm on the shaking machine. The solutions were then membrane filtered (0.45 µm) before 600 µL of the filtrate was transferred to an NMR tube followed by analysis.

#### 2.2.2. NMR Analysis

Two 400 MHz (9.4 T) field strength spectrometers were used to acquire proton NMR spectra: an AVANCE 400 Ultra Shield with a 5 mm PASEI 1H/D-13C Z-GRD probe, and an Ascend 400 with a BBI 400S1 H-BB-D-05 Z (each from Bruker, Rheinstetten, Germany). All samples were measured in 5 mm sample tubes (NMR tube DEU-Quant 5 mm, 7 inch) (Deutero, Kastellaun, Germany). The spectra were automatically acquired at 300.0 K under the control of Sample Track and ICON-NMR (Bruker BioSpin, Rheinstetten, Germany). Detailed information about measurement methodology is available in [26].

A waiting time of 5 min for temperature equilibration was used for every measurement. The NMR spectra were acquired using a Bruker pulse program (zg30) with 64 scans (NS) and 2 prior dummy scans (DS) with a relaxation delay (D1) of 30 s and an acquisition time of 7.97 s. The time domain was set to 131072 data points with a spectral width of 20.5503 ppm (8223.68 Hz) for UltraShield and 20.5617 ppm (8223.69 Hz) for Ascend. The size of the real spectrum (SI) was 262144. The receiver gain was set to 45.2. All spectra were recorded with the basopt mode. The acquisition parameters were constant for all spectra for pulse length–based concentration determination (PULCON, see Wider & Dreier [33]) measurement according to Lachenmeier et al. [26]. The free induction decay (FID) was multiplied with an exponential window function to achieve a line broadening of 0.30 Hz. The spectra were automatically phased and baseline-corrected (default settings) using TopSpin version 3.2 and 3.5 (Bruker Biospin, Rheinstetten, Germany).

#### 2.2.3. Experimental Design

A factorial experimental design was adopted for the validation studies. For this purpose, six matrix calibration series, each consisting of two blanks and ten samples with increasing amounts namely 1, 5, 10, 25, 50, 100, 250, 500, 750, and 1000 mg/L of analyte were prepared. A factorial design was employed for the investigation of the influence of the three experimental factors, NMR spectrometer type, coffee type and shaking time (see Table 2). Each measurement series corresponds to a different combination of factor characteristics (see Appendix A for full design).

#### 2.2.4. Preparation of Working and Test Solutions

Stock solutions (1000 and 5000 mg/L) of each of the analytes comprising caffeine, HMF, OMC, kahweol, and furfuryl alcohol were used to prepare 1, 5, 10, 25, 50, 75, 125, 250, 500, 750, and 1000 mg/L working solutions. Additionally, two blanks were made for each of the measurement series. Separate test solutions were prepared for the three matrices (100% arabica decaffeinated, 100% robusta coffee and green coffee). The dilution matrix to achieve the desired concentration is shown in Table 3.

Therefore, for the six matrix calibration series, a total of 72 test solutions was prepared. However, since all samples were run in two instruments (Ultrashield/Ascend), 144 measurement results were obtained (or 120 without the blank values).

### 2.3. Validation Studies

Three different coffee matrices spanning the broadest possible spectrum of different coffee constituents were used during validation. These consisted of decaffeinated coffee (decaf. arabica, matrix 1), robusta coffee (matrix 2), and raw coffee (green coffee, matrix 3). For the preparation of the spiked matrix samples, each pure substance was weighed before being dissolved in CDCl_3_ and TMS solution (usually in 5–10 mL). Subsequently, the test samples were spiked in the specified concentration range (see Appendix A). The control solution was also run after a series of measurements in order to ascertain that analyses were properly performed so that test results obtained could be considered reliable.

#### 2.3.1. Selectivity

The selectivity of each analyte was established by measurement of all analytes mixed in a solution without matrix. To achieve this, 100 µL of each of the five analytes were pipetted into a NMR tube followed by addition of 500 µL CDCl_3_ before NMR analysis (desired concentration 500 mg/L).

Furthermore, all analytes in a solution were mixed with 100% arabica decaffeinated coffee in order to check possible matrix disturbances. This was achieved by pipetting 150 µL of each of the five analytes into an NMR tube followed by the addition of 750 µL CDCl_3_, before adding 200 mg of coffee sample (desired concentration 100 mg/L). The solution was shaken for 20 min at 350 rpm on the shaker, then membrane filtered and used directly for NMR measurement. For comparison, an NMR spectrum of a coffee sample (without analytes) was also acquired. The coffee sample was prepared by dissolving 200 mg coffee sample in 1.5 mL CDCl_3_, shaken for 20 min at 350 rpm on the shaker, membrane filtered, and used for NMR analysis.

#### 2.3.2. Detection and Quantification Limits

In order to determine the detection limit, 9 spiking levels of different concentrations were added to decaffeinated arabica coffee (Matrix 1), processed, measured and evaluated. The detection and determination limits were determined in the lower working range according to the German norm DIN 32645 [34].

#### 2.3.3. Precision and Recovery

For the determination of the measurement uncertainties and the recoveries, 9 spiking levels at different concentrations were added to the 3 matrices and processed. The measurement uncertainty was determined with the aid of ANOVA (all settings at default) using Design Expert Software V.7.0 (Stat-Ease Inc., Minneapolis, MN, USA).

### 2.4. Data Analysis and Quality Control

Peak areas in the 1D-proton NMR spectra were evaluated with the help of a compiled MatLab script. The peak areas were determined using a line fitting algorithm. Quantification was performed using the eretic factor, which was previously determined using a quant reference (for details see [26]). At the end of each measurement series, the control solution was measured as a safeguard. The assignment of the signal patterns and the determination of the exact position of the signals were performed by the analysis of a 2D-JRES-NMR spectrum. Note: due to the restricted solubility of caffeine in CDCl_3_, an empirical factor of 6 for recalculation has to be used, as determined based on HPLC measurements using the German reference procedure [35].

### 2.5. Method Performance

The method was assessed for performance by calculating the standard deviation of the intra- laboratory reproducibility, recovery, robustness, limits of measurements, and the total uncertainty of the measurements as a function of concentration.

## 3. Results and Discussion

### 3.1. Validation

#### 3.1.1. Specificity and Selectivity

The use of working reference standards enabled accurate assignment of chemical shifts. The chemical shifts of the analytes in the different matrices are shown in Table 4. A representative spectrum of an authentic sample including magnifications of target resonances is provided in Figure 1. OMC presented a slight offset in the integration range that led to a too high integral due to matrix interferences (especially fatty acids in the field region higher to OMC). However, this problem was circumvented by integrating the range next to OMC (3.04–3.10 ppm), which is similarly affected by the same matrix interference, and subtracting it from the sum of the integral of OMC (also see an illustration of the problem in Figure 2).

#### 3.1.2. Analytical Limits

The detection and quantification limits are given in Table 5 together with the concentration ranges. The limits of measurements were adjudged fit for purpose.

#### 3.1.3. Precision

The recoveries of the different analytes in various matrices are shown in Table 6. Although, the recoveries in a majority of the matrices used were within limits, green coffee gave poor unsatisfactory recoveries for caffeine, OMC and kahweol. Moreover, the recovery of kahweol from robusta coffee was out of specifications too (see Table 6).

The coefficient of variation (CV) was used as criterion for evaluating the precision of the proposed NMR method. The acceptance criterion for precision was a CV of less than 15% (internal quality standard of the authors’ laboratory). Apart from kahweol, the precision of all the other analytes was found being within the limits of acceptance in all matrices. The analytes, caffeine, OMC, furfuryl alcohol and HMF present in roasted coffee (arabica and robusta) can therefore be determined with sufficient precision and accuracy by using the proposed NMR method. However, kahweol may not be quantified with adequate precision in robusta due to its low content in this matrix. In addition to the out-of-specification recoveries, the precision of all the analytes for green coffee were unsuitable (Table 7). Further work, potentially by improving the extraction, appears to be necessary for green coffee.

#### 3.1.4. Linearity of Detector Response

Linearity was established in the concentration ranges (working range) listed in Table 8. The linearity was determined in matrix 1. Since the coefficients of determination (R^2^) were all >0.99 over the concentration ranges examined, the method may be considered to be fit-for-purpose.

#### 3.1.5. Effect of Matrix

ANOVA revealed that the models are significant for all analytes and can be evaluated. For all analytes it was shown that the instrument used (i.e., NMR spectrometer type) has no significant influence on the analytical results. The measurements can thus be performed on both spectrometers. Similarly, the extraction time had no significant influence. If the results are viewed manually, the extraction time of 20 min seems adequate, but not statistically significant, to achieve better results, and was thus defined as a setting.

However, the influence of variety of coffee was found to be statistically significant especially with green coffee, which had a significantly greater dispersion. Roasting was found to have no influence on the determinations since similar recoveries were obtained for the analytes. The method can therefore only be considered successfully validated for the determination of OMC, caffeine, kahweol, furfuryl alcohol, and HMF in samples of roasted coffee. Measurements of green coffee shall be considered as indicative only.

#### 3.1.6. Applicability

The method was applied to 797 samples since 2016. Suspicious samples, i.e., cases of potential food fraud (arabica samples with OMC > 50 mg/kg) were in all cases positively confirmed using the German norm procedure based on HPLC [35]. Furthermore, the applicability of the method was proven during the recent OPSON VIII Europol-Interpol operation [36,37], in which more than 150 roasted coffee samples were analyzed using the validated NMR procedure within the two-week operation period (see, e.g., [36]). In this sample, three cases of substantial admixture of robusta into coffee claimed as 100% arabica could be determined.

## 4. Conclusions

The proposed NMR spectroscopic method gave satisfactory validation results for specificity, selectivity and linearity. All analytes examined gave satisfactory recoveries except caffeine, OMC and kahweol in green coffee and kahweol in robusta coffee (due to its very low content in this matrix). The analytical limits were found to be adequate for routine NMR measurements for the analytes. Importantly, the proton NMR spectroscopic method was found to be suitable for unambiguously coffee screening and authenticity testing. Additionally, the method may be adopted for the routine quantitation of furfuryl alcohol in coffee in analytical laboratories.

## Figures and Tables

**Figure 1 foods-09-00047-f001:**
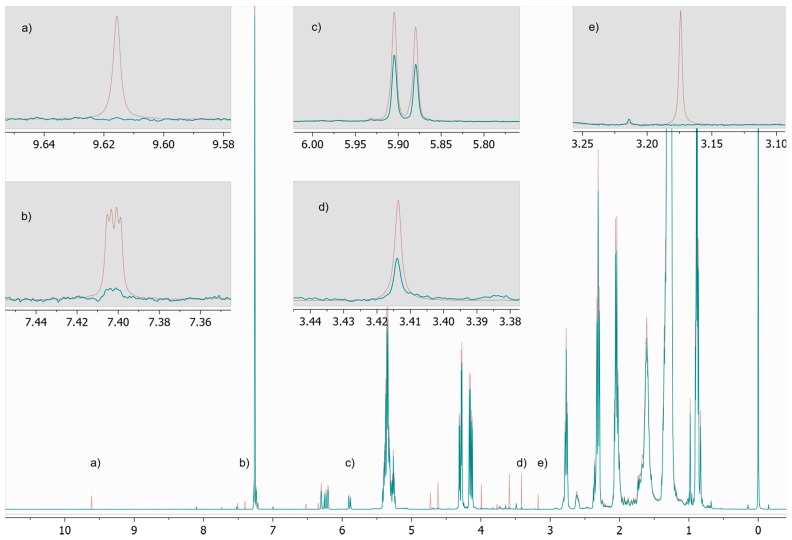
Representative 1H NMR spectrum of an authentic coffee sample showing target resonances in magnification (HMF (**a**), furfuryl alcohol (**b**), kahweol (**c**), caffeine (**d**), OMC (**e**)).

**Figure 2 foods-09-00047-f002:**
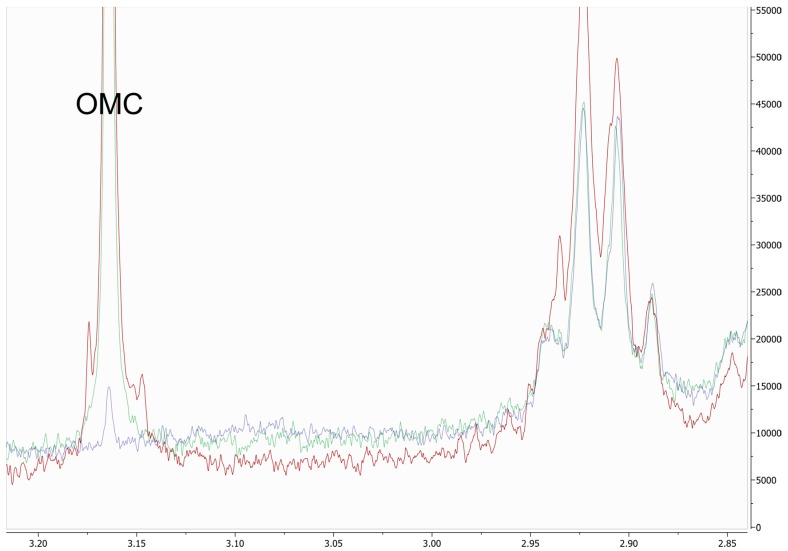
NMR range besides the OMC signal showing matrix interferences from resonances of fatty acid signals around 2.9 ppm. Over-quantification can be avoided by subtracting the noise range (3.04 ppm–3.10 ppm).

**Table 1 foods-09-00047-t001:** Working standards used.

Substance	Guidance Value ^a^ (mg/kg)	Defined Working Ranges According to Experience (mg/kg)
OMC	<50 for arabica	7.5–7500
Caffeine	<1000 for decaf	7.5–7500
Kahweol	<300 for robusta	7.5–7500
Furfuryl alcohol	-	7.5–7500
HMF	-	7.5–7500

^a^ Guidance for OMC and kahweol based on own experience in analyzing coffee samples. Guidance for caffeine in decaf coffee is the limit in German national coffee regulation (“Kaffeeverordnung”) [32].

**Table 2 foods-09-00047-t002:** Factorial experimental design used.

Array	Factor 1: NMR Device	Factor 2: Coffee Type	Factor 3: Shaking Time (min)
1	Ultrashield/Ascend	100% arabica decaffeinated	20
2	Ultrashield/Ascend	100% robusta	20
3	Ultrashield/Ascend	Green coffee	20
4	Ultrashield/Ascend	100% arabica decaffeinated	10
5	Ultrashield/Ascend	100% robusta	10
6	Ultrashield/Ascend	Green coffee	10

**Table 3 foods-09-00047-t003:** Dilution matrix.

Desired Calibration Concentration (mg/L)	Dilution of Stock Solution (1000 mg/L)	Desired Final Concentration (mg/L)	Volume per Stock Solution (µL)	Volume of all Analytes (µL)	Volume of CDCl_3_ (µL)
0 (Blank, 2×)	0	0	-	-	1500
1	1:1000	1	1.5	7.5	1492.5
5	1:200	5	7.5	37.5	1462.5
10	1:100	10	15	75	1425
25	1:40	25	37.5	187.5	1312.5
50	1:20	50	75	375	1125
100	1:10	100	150	750	750
250	1:20	250	75	375	1125
500	1:10	500	150	750	750
750	1:6.66	750	225	1125	375
1000	1:5	1000	300	1500	-

**Table 4 foods-09-00047-t004:** Characteristic signals of the constituents in coffee and their ranges.

Analyte	Integration Range (ppm)
OMC	3.185–3.125
Caffeine	3.44–3.38
Kahweol	5.925–5.85
Furfuryl alcohol	7.411–7.39
HMF	9.69–9.67

**Table 5 foods-09-00047-t005:** Limits of detection and quantification of analytes determined.

Analyte	Detection Limit (mg/kg)	Determination Limit (mg/kg)	Concentration Range for Determination of Limit (mg/kg)
OMC	2.5	7.4	7.5–187.5
Caffeine	15.7	43.1	7.5–187.5
Kahweol	186.0	501.4	187.5–1875.0
Furfuryl alcohol	11.6	39.4	7.5–75
HMF	6.3	22.9	7.5–75

**Table 6 foods-09-00047-t006:** Recovery of coffee constituents from different matrices.

Matrix	Recovery (%)				
	Caffeine	OMC	Kahweol	Furfuryl Alcohol	HMF
Decaf. arabica	101	97	95	97	102
Robusta	102	101	74 *	99	101
Green coffee	137 *	54 *	188 *	93	107

* Outside of specification. Specification: 90–110%.

**Table 7 foods-09-00047-t007:** Precision of coffee constituents.

Matrix	Precision (CV)				
	Caffeine	OMC	Kahweol	Furfuryl Alcohol	HMF
Decaf. arabica	8.1	6.5	22.2 *	6.1	8.3
Robusta	7.4	7.8	32.7 *	5.8	6.9
Green coffee	104 *	188 *	570 *	25 *	27 *

* Outside of specification. Specification: <15%. CV—coefficient of variation.

**Table 8 foods-09-00047-t008:** Linear concentration range of the coffee analytes (also see Appendix A).

Analyte	Linearity (mg/kg)	Coefficient of Determination (*R*^2^)
OMC	7.5–5625	1.0000
Caffeine	7.5–5625	1.0000
Kahweol	7.5–5625	0.9949
Furfuryl alcohol	7.5–5625	1.0000
HMF	7.5–5625	0.9997

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
