# Peer review of "Validation of a Quantitative Proton Nuclear Magnetic Resonance Spectroscopic Screening Method for Coffee Quality and Authenticity (NMR Coffee Screener)"

_foods, 2020, doi:10.3390/foods9010047_

Round 1

Reviewer 1 Report

In this work the authors tried to validate an analytical method based on quantitative NMR spectroscopy, proposed in other previously published work, to monitor the quality and the authenticity of the coffee and prevent fraud. The subject of the work is very interesting from the commercial point of view but the development of the methodology used to validate the method seems cumbersome and not easy to follow. In order to improve the quality and the readability of the manuscript my suggestion is to modify some parts trying to make them clearer and more usable. In particular the authors should consider the comments below.

In the section 2.2.1. “ Samples and sample preparation”, line 82 “Analyte sample ……” what analytes are they? real coffee samples? it is not clear to which analytes they refer. The authors should specify which analytes are and why they weigh 200 mg, that for a routine NMR spectrum is a large amount. Section 2.2.2 “ NMR analysis”: on which magnetically active nucleus were NMR spectra recorded? The authors throughout the manuscript not say which nuclei  have been taken into consideration until the section 2.4 and which spectra have been recorded.  Only in section 2.4 do they indicate that spectra are the proton spectrum, but the authors should  specify it first  and also should reported at least one of these spectra in the paper. Section 2.2.2 “ NMR analysis”, lines 90 and 93 in both is reported the temperature of 300 °C. Section 2.2.2 “ NMR analysis”: a)line 94, the authors report a D1 of 30 s. Why a D1 so long? Generally  for the 1H it is much shorter; b) line 95 “… with a spectral width of 20.5503…..” should it be in ppm? The same at line 96. Section 2.3.1. “Selectivity”, line 132, the authors said: “…… parameters namely coupling constant, multiplicity ….” where are these parameters shown? Line 133: what are the five analytes?   The whole section should be written more clearly.

Author Response

In this work the authors tried to validate an analytical method based on quantitative NMR spectroscopy, proposed in other previously published work, to monitor the quality and the authenticity of the coffee and prevent fraud. The subject of the work is very interesting from the commercial point of view but the development of the methodology used to validate the method seems cumbersome and not easy to follow. In order to improve the quality and the readability of the manuscript my suggestion is to modify some parts trying to make them clearer and more usable. In particular the authors should consider the comments below.

Response: Thank you for your advice.

In the section 2.2.1. “Samples and sample preparation”, line 82 “Analyte sample ……” what analytes are they? Real coffee samples? It is not clear to which analytes they refer. The authors should specify which analytes are and why they weigh 200 mg that for a routine NMR spectrum is a large amount.

Response: Analyte samples refer to authentic coffee samples from the market. You need 200 mg sample weight because the actual analytes are contained in much lower concentrations, such as less than 50 ppm for 16-OMC. The section was clarified as requested.

Section 2.2.2 “NMR analysis”: on which magnetically active nucleus were NMR spectra recorded? The authors throughout the manuscript not say which nuclei have been taken into consideration until the section 2.4 and which spectra have been recorded.  Only in section 2.4 do they indicate that spectra are the proton spectrum, but the authors should specify it first and also should report at least one of these spectra in the paper.

Response: Actually the title already specifies proton (=1H) NMR. As requested the information about proton NMR was more prominently added to the text throughout and a figure of spectra was added as requested (see also responses to requests below).

Section 2.2.2 “NMR analysis”, lines 90 and 93 in both is reported the temperature of 300 °C.

Response: The double information was deleted from line 93.

Section 2.2.2 “NMR analysis”: a) line 94, the authors report a D1 of 30 s. Why a D1 so long? Generally for the 1H it is much shorter;

Response: For furfuryl alcohol there is the longest relaxation delay of 6 s T1. This means that we have to use 5x times 6 s = 30s D1 for safety reasons.  The 30° pulse is used because we have a shorter measurement time. For 90° pulse we would have an even longer relaxation delay.

b) line 95 “… with a spectral width of 20.5503…..” should it be in ppm? The same at line 96.

Response: The units were added.

Section 2.3.1. “Selectivity”, line 132, the authors said: “…… parameters namely coupling constant, multiplicity ….” where are these parameters shown?

Response: The sentence was revised. Selectivity was ensured by comparing the spectra of reference substances measured without matrix.

Line 133: what are the five analytes?   The whole section should be written more clearly.

Response: The five analytes are now spelled out directly at the beginning of the methods section.

Reviewer 2 Report

The manuscript by Okaru et al. developed and validated an NMR based method to assess the coffee quality and authenticity. Overall the experiments presented are well controlled, although the findings are not particularly extensive and innovative. Here below are listed my minor comments:

1). Please provide some NMR raw spectrum as references, which will help improve the overall quality of this manuscript.

2). It will be better if the linearity of detection can be shown as plots in a figure. The R square is 1.0000, which is a little bit surprising to me.

Author Response

The manuscript by Okaru et al. developed and validated an NMR based method to assess the coffee quality and authenticity. Overall the experiments presented are well controlled, although the findings are not particularly extensive and innovative. Here below are listed my minor comments:

Response: Thank you for your helpful comments.

1). Please provide some NMR raw spectrum as references, which will help improve the overall quality of this manuscript.

Response: Spectra are now provided; see also comments to other reviewers.

2). It will be better if the linearity of detection can be shown as plots in a figure. The R square is 1.0000, which is a little bit surprising to me.

Response: This may be the results of the excellent linear detector response of NMR along with rounding to four digits. As we believe that showing all the linear relationships may be excessive, we now show the figures in an appendix but not in the main body.

Reviewer 3 Report

This manuscript presents a validated qNMR method for the purpose of screening the quality and authenticity of coffee. Working reference standards were used to establish the analytical limits of the method. A factorial experimental design for analyzing various coffee was then implemented. The basic set-up of the paper seems well-thought out for an important problem. Indeed, NMR is an ideal tool for this problem.

Despite the potential positive impact of this study, there is a number of shortcomings. Taking my comments collectively, it is highly questionable if this validated method could be repeated by another laboratory with the information given.

In Table 1, what are the guidance values? Where did these come from? How did you establish the working ranges? Did these come from regulatory documents? Or literature values?  In section 2.2.1, what was the origin of the coffee?

     3. In section 2.2.2, the NMR method appears to draw heavily from reference 26. However, neither I nor the editor could not find this article. What is the PULCON method? This is never described. Especially since reference 26 is not available as of this review, all the methods relevant to the validated method need to be described.

     4. More detail is needed on the processing of the NMR data, as this can affect the results. How was the exponential weighting function applied? There are different baseline corrections algorithms within Topspin. What one was used.

      5. 131072 is the number of points. Please add the unit.

      6. In section 2.2.3, discuss Table SI a little more, either in the section or in the table caption. In particular, what is factor 1? What is being spiked in? All three working standards?

      7. In section 2.3, what is the control? Is this the blank? If so, make sure the terminology is consistent. Otherwise, state precisely what the control is.

       8. In section 2.3.3, how was the ANOVA software used? What were the settings or software flags? As this is a validated method, this information is important.

       9. In section 2.4, again, what was the control? Also, what is the German reference procedure? From DIN32645?

       10. In section 3.1.3, an acceptance criterion was set at less than 15% for the CV. How was this number determined?

While data is extracted from NMR spectra, none are actually given to access spectral quality. Several representative 1D overlays are suggested. In the figure, show the integration ranges. Secondly, in Section 3.1.1 no justification is given for circumventing the OMC integration problem. Indeed, treatment of the integrals in this way is not appropriate, unless this same ‘blank’ integration region is subtracted from all integration regions. An investigation needs to be taken into why the OMC values are high. Are there other signals under OMC? Maybe some other contributing factor? As is, the OMC values need to be considered an outlier.

Lastly, the manuscript ends with 2 sentences describing the application of this validated method to a real-world problem. Such an application is very appropriate to show that the method is working as expected. However, this should be a separate data section in and of itself. The data needs to be provided, so that the reader can evaluate how well the method performed.

Author Response

This manuscript presents a validated qNMR method for the purpose of screening the quality and authenticity of coffee. Working reference standards were used to establish the analytical limits of the method. A factorial experimental design for analyzing various coffee was then implemented. The basic set-up of the paper seems well-thought out for an important problem. Indeed, NMR is an ideal tool for this problem.

Response: Thank you.

Despite the potential positive impact of this study, there is a number of shortcomings. Taking my comments collectively, it is highly questionable if this validated method could be repeated by another laboratory with the information given.

Response: We believe that all parameters are provided making the methods replicable. It should be kept in mind that method development and optimization has been reported in another paper, see comment below.

In Table 1, what are the guidance values? Where did these come from? How did you establish the working ranges? Did these come from regulatory documents? Or literature values? 

Response: A footnote about the guidance values is now provided. The working range was defined by us based on our experience as already stated in the column heading.

In section 2.2.1, what was the origin of the coffee?

Response: Information about sample origin was added.

In section 2.2.2, the NMR method appears to draw heavily from reference 26. However, neither I nor the editor could not find this article. What is the PULCON method? This is never described. Especially since reference 26 is not available as of this review, all the methods relevant to the validated method need to be described.

Response: The paper is currently in proof stage at J. AOAC and should appear soon (http://dx.doi.org/10.1093/jaocint/qsz020). For information of the reviewers and the editor, we have uploaded the paper as well as the acceptance letter as supporting information (not for publication) in your manuscript system. The PULCON method is a common principle for NMR quantification, the abbreviation is now spelled out. Three further references were added to offer more background on qNMR, PULCON etc. 

More detail is needed on the processing of the NMR data, as this can affect the results. How was the exponential weighting function applied? There are different baseline corrections algorithms within Topspin. What one was used.

Response: The details are now provided in the text as requested.

131072 is the number of points. Please add the unit.

Response: These are data points (number without unit).

In section 2.2.3, discuss Table SI a little more, either in the section or in the table caption. In particular, what is factor 1? What is being spiked in? All three working standards?

Response: Yes, this information was added to the table header in column 1.

In section 2.3, what is the control? Is this the blank? If so, make sure the terminology is consistent. Otherwise, state precisely what the control is.

Response: Actually, it was no blank solution, but a control solution with defined content of a substance. The missing information was added to the text.

In section 2.3.3, how was the ANOVA software used? What were the settings or software flags? As this is a validated method, this information is important.

Response: All software settings were at default. I believe that there actually are no triggers in the ANOVA algorithm of Design expert, which would lead to different results. It should also be noted that this was only used to evaluate the experimental results and does not influence the actual validation.

In section 2.4, again, what was the control? Also, what is the German reference procedure? From DIN32645?

Response: Yes, DIN 32645 is a German norm procedure. A reference to the standard was added to the reference list (#30).

In section 3.1.3, an acceptance criterion was set at less than 15% for the CV. How was this number determined?

Response:  The rationale was added to the text (internal arbitrary quality criterion in our lab).

While data is extracted from NMR spectra, none are actually given to access spectral quality. Several representative 1D overlays are suggested. In the figure, show the integration ranges.

Response: The figures were compiled and are now provided as requested.

Secondly, in Section 3.1.1 no justification is given for circumventing the OMC integration problem. Indeed, treatment of the integrals in this way is not appropriate, unless this same ‘blank’ integration region is subtracted from all integration regions. An investigation needs to be taken into why the OMC values are high. Are there other signals under OMC? Maybe some other contributing factor? As is, the OMC values need to be considered an outlier.

Response: The offset is only affecting the OMC signal and none of the other signals (see figures). OMC is also the smallest of the signal. To better illustrate the problem, a new figure was compiled. It can be seen that matrix interferes the whole region where OMC is located. To avoid over-quantification, we decided to subtract the noise offset from the OMC signal integral. The explanation was improved as requested and an additional figure compiled which shows the problem. As our applicability data prove, this pragmatic approach to overcome the problem is working absolutely fine and does not make the values to outliers.

Lastly, the manuscript ends with 2 sentences describing the application of this validated method to a real-world problem. Such an application is very appropriate to show that the method is working as expected. However, this should be a separate data section in and of itself. The data needs to be provided, so that the reader can evaluate how well the method performed.

Response: The text was moved in a separate section as requested and expanded also providing references to results. The raw data cannot be provided for confidentiality reasons (criminal cases in progress).

Round 2

Reviewer 1 Report

The authors replied to all comments, therefore the manuscript can now be accepted for publication in this last version.